# Postnatal Changes of Neural Stem Cells in the Mammalian Auditory Cortex

**DOI:** 10.3390/ijms22041550

**Published:** 2021-02-04

**Authors:** Zhengqing Hu, Li Tao, Meng Deng

**Affiliations:** 1John D. Dingell VA Medical Center, Detroit, MI 48201, USA; 2Department of Otolaryngology-Head and Neck Surgery, Wayne State University School of Medicine, Detroit, MI 48201, USA; ltao@wayne.edu (L.T.); april@wayne.edu (M.D.)

**Keywords:** auditory cortex, development, neural stem cell, neurosphere, postnatal, proliferation

## Abstract

Our previous study reported neural stem cells (NSCs) in the auditory cortex (AC) of postnatal day 3 (P3) mice in vitro. It is unclear whether AC-NSCs exist in vivo. This study aims to determine the presence and changes of AC-NSCs during postnatal development and maturation both in vitro and in vivo. P3, postnatal day 14 (P14), 2-month-old (2M), and 4-month-old (4M) mouse brain tissues were fixed and cryosectioned for NSC marker immunostaining. In vitro, P3, P14, and 2M AC tissues were dissected and cultured in suspension to study NSCs. NSC proliferation was examined by EdU incorporation and cell doubling time assays in vitro. The results show that Nestin and Sox2 double expressing NSCs were observed in the AC area from P3 to 4M in vivo, in which the number of NSCs remarkably reduced with age. In vitro, the neurosphere forming capability, cell proliferation, and percentage of Nestin and Sox2 double expressing NSCs significantly diminished with age. These results suggest that AC-NSCs exist in the mouse AC area both in vitro and in vivo, and the percentage of AC-NSCs decreases during postnatal development and maturation. The results may provide important cues for the future research of the central auditory system.

## 1. Introduction

In the auditory system, the sound is captured and converted into auditory signals in the ear, which is transferred to the central auditory system, including the auditory cortex (AC). The peripheral auditory system is vulnerable to a variety of insults, including sound overstimulation, genetic disorder, ototoxic drugs, and trauma, which usually leads to irreversible damage to hair cells and spiral ganglion neurons to cause hearing impairment [1,2,3]. Extensive efforts have been focused on the regeneration of the peripheral auditory system, including hearing aids, cochlear implants, stem cell-based replacement, gene therapy, and stria vascularis regeneration [4,5,6,7,8,9,10,11,12,13]. In the meantime, the central auditory system may be injured by primary insults, including infection, tumor, and head injury [14,15,16]. The central auditory system can also be damaged by degeneration secondary to the peripheral auditory system injury. It is currently unclear whether the central auditory system can be regenerated following injuries, which is an understudied area.

A possible approach to remedy the central auditory system degeneration is the application of stem cells. The stem cell approach may be most effective for head trauma involving the temporal bone and advanced auditory degeneration cases, in which a significant number of central auditory cellular structures are damaged. To test the possibility of the stem cell-based approach, it is imperative to understand whether stem cells exist in the central auditory system. It is known that stem cells exist in the adult mammalian subventricular zone of the lateral ventricles and the subgranular zone of the hippocampus [17,18,19,20]. The neural stem cells (NSCs) of these two regions express NSC proteins, including Sox2 and Nestin [21,22]. Additionally, subventricular and subgranular NSCs can proliferate and differentiate into neurons both in vitro and in vivo [17,19,21,23,24]. In the central auditory system, a previous study reported the presence of NSCs in the cochlear nucleus, which possesses the capability to proliferate and differentiate into neural cells in vitro [25]. However, the report of the AC stem cell research is very limited.

In our previous report, it was observed that postnatal day 3 (P3) mouse AC cells expressed NSC proteins Sox2 and Nestin, which were able to proliferate and differentiate into cells expressing neuronal markers beta III tubulin, neurofilament light chain, and NeuroD1 in vitro [26]. This study suggests the existence of NSCs in postnatal mouse AC in vitro. However, a few fundamental questions have not been addressed. For instance, it is unclear whether the fate of the AC NSC is altered during postnatal development and maturation. It has not been determined whether AC NSCs exist in vivo. Therefore, the current study aims to determine whether AC NSCs exist in vivo and to define the changes of AC NSCs during postnatal development using a mouse model.

## 2. Results

### 2.1. Identification of AC-NSCs In Vivo

Our previous reports showed that the AC tissue was harvested from the postnatal mouse brain, and AC-NSCs were identified in the culturing of these tissues [26]. However, it remains unclear whether NSCs exist in the mouse AC tissue in vivo. To address this question, cryosections of postnatal day 3 (P3), postnatal day 14 (P14), 2-month-old (2M), and 4-month-old (4M) mouse brain were labeled with Hoechst 33342, anti-Nestin, and anti-Sox2 antibodies (Figure 1A,B). In this study, Sox2 (nuclei protein) and Nestin (intermediate filament protein) double-labeled cells were considered as NSCs [27,28,29]. It was found that a proportion of cells located in the AC area expressed NSC proteins, including Nestin and Sox2, on P3, P14, 2M, and 4M mouse AC cryosections (Figure 1B), suggesting the presence of NSCs in the AC at different postnatal and adult ages (Figure 1B,C).

In the quantitative study, the percentages of Nestin positive, Sox2 positive, and double-positive cells were 43.8 ± 6.5%, 16.8 ± 5.1%, and 15.4 ± 4.6% in the P3 AC, 30.1 ± 2.3%, 15.5 ± 2.5%, and 5.5 ± 1.1% in the P14 AC, 29.5 ± 2.6%, 18.7 ± 3.0%, and 3.7 ± 0.9% in the 2M AC, and 9.2 ± 2.4%, 12.1 ± 1.0%, and 1.6 ± 0.1% in the 4M AC, respectively (mean ± standard error; Figure 1C). It was observed that the percentage of Nestin positive cells of the P3, P14, and 2M AC was significantly higher than the 4M AC (*p* < 0.01, Analysis of variance (ANOVA), n = 5 mice in each group), whereas there was no significant difference among the P3, P14, and 2M groups (*p* > 0.05, ANOVA and Tukey tests). The percentage of Sox2 positive cells was not significantly different among these 4 groups (*p* > 0.05, ANOVA). The percentage of Nestin and Sox2 double-positive cells of the P3 group was significantly higher than those of the P14, 2M, and 4M groups (*p* < 0.05 for P3 vs. P14 and P3 vs. 2M; *p* < 0.01 for P3 vs. 4M; ANOVA followed by Tukey post hoc tests). No significant difference was identified among the P14, 2M, and 4M groups (*p* > 0.05, ANOVA followed by Tukey post hoc test; Figure 1C).

Taken together, these data suggest that Nestin and Sox2 double expressing cells exist in the P3, P14, 2M, and 4M mouse AC in vivo. The number of AC NSCs decreases during postnatal development and maturation.

### 2.2. Generation of Spheres Using P3, P14, and 2M Mouse AC Tissues

Our previous in vitro study suggests the presence of AC-NSCs in the P3 mouse AC [26]. However, it remains unclear whether it is possible to identify and culture NSCs from mature AC tissues. To address this issue, AC tissues of P3, 14, and 2M mice were harvested and cultured for 3 passages in suspension using our previously published methods [26,27,28,30]. The AC tissues and derived spheres were dissociated into singular cells, which were cultured in the suspension medium for 5–7 days for spherical structure formation. Primary and secondary cell clusters and spherical structures were observed in all 3 groups (Figure 2). Tertiary spheres were only found in the P3 and P14 groups, whereas they were very small and rarely seen in the 2M group. Since the distinct spherical structures were usually formed in the tertiary spheres, the tertiary spheres were analyzed in this study. It was found that the size of the spheres was different among groups. In the quantitative study, the diameter of the tertiary spheres was 165.2 ± 13.9 µm and 86.6 ± 6.0 µm for the P3 and P14 groups respectively (mean ± standard error; *p* < 0.01, n = 12 culture samples from 3 independent primary cultures, Student’s *t*-test; Figure 2). The tertiary spheres were hardly found in the 2M group, which excluded them from quantification and the following in vitro studies.

### 2.3. Proliferation of P3 and P14 AC Spheres In Vitro

Complementary cell doubling time and EdU incorporation assays were used to evaluate the proliferation of newly generated tertiary spheres of the P3 and P14 groups. It was found that the cell doubling time of the P3 group was 36.4 ± 2.5 h, whereas it was 108.9 ± 13.6 h in the P14 group. The statistical analysis showed a significant difference (*p* < 0.01, n = 7 culture samples from 3 independent primary cultures, Student’s *t*-test).

In the EdU proliferation study, EdU was added to the culture medium during culture days 2–4 for 48 h in the tertiary culture of the P3 and P14 groups. The samples were fixed at the end of the culture experiment. It was observed that the percentages of EdU incorporated cells were 31.7 ± 2.8% and 9.3 ± 1.6% in the P3 and P14 groups, respectively. Statistical analysis suggests a significant difference (*p* < 0.01, n = 10 culture samples from 3 independent primary cultures, Student’s *t*-test; Figure 3). These data suggest that NSCs of both P3 and P14 groups can proliferate, and the proliferation capability reduces with age.

### 2.4. Expression of NSC Proteins in AC-NSCs In Vitro

Our previous study shows that P3 AC cells expressed NSC proteins Nestin and Sox2 in vitro [26]. To determine the expression of NSC proteins in the late postnatal stages, Nestin and Sox2 immunostaining was applied to the P3 and P14 AC-derived tertiary spheres. It was found that NSC proteins Nestin and Sox2 were expressed in both groups. The percentages of Nestin positive, Sox2 positive, and double-labeled cells were 84.6 ± 6.1%, 88.6 ± 2.3%, and 81.2 ± 5.2%, respectively, in the P3 group, whereas the numbers were 43.9 ± 3.8%, 77.6 ± 4.4%, and 41.6 ± 4.4%, respectively, in the P14 group (mean ± standard error; Figure 4). The percentages of Nestin positive, Sox2 positive, and double-positive cells of the P3 group are significantly higher than the P14 group (*p* < 0.05 or *p* < 0.01, Student’s *t*-test, n = 10 culture samples from 3 independent primary cultures; Figure 4). This study suggests that NSC proteins were expressed in the AC area of both P3 and P14 groups.

## 3. Discussion

In our previous report, AC-NSCs were identified in the P3 mouse brain, and these NSCs were able to be cultured in vitro [26]. In this study, P3, P14, 2M, and 4M mouse AC tissues were studied for the presence of NSCs in vivo. The NSC markers Sox2 and Nestin double-positive cells were found in the mouse AC sections from P3 to 4M, suggesting the presence of NSCs in the mouse AC area in vivo. In the in vitro study, it was found that AC NSCs of the P3 and P14 groups were able to proliferate, form neurospheres for at least 3 passages, and express NSC proteins Sox2 and Nestin.

In the in vivo study, it was found that NSC proteins Nestin and Sox2 were expressed in the P3, P14, 2M, and 4M AC sections. Nestin is an intermediate filament protein that is usually observed in the nervous tissue and NSCs [31,32]. Sox2 is a transcription factor that plays an important role in maintaining the stem cell features of embryonic stem cells and neural stem cells. Sox2 also has an essential downstream role in the differentiation of specific neuron subtypes [33,34]. Simultaneous expression of Nestin and Sox2 usually indicates the presence of NSCs [29,30,35]. In this study, Nestin and Sox2 immunostaining was observed in the AC tissues of all 4 groups, including P3, P14, 2M, and 4M. The percentages of Sox2-expressing cells were similar among 4 groups, approximately 17%, 15%, 19%, and 12% for P3, P14, 2M, and 4M groups respectively (*p* > 0.05, ANOVA). However, the percentage of Nestin-expressing cells decreased significantly with age, approximately 44%, 30%, 30%, and 9% for P3, P14, 2M, and 4M groups, respectively. Similarly, the percentage of Nestin and Sox2 double expressing cells significantly diminished with age, approximately 15%, 5%, 4%, and 2% for P3, P14, 2M, and 4M groups, respectively. These data suggest that the percentage of AC NSCs reduces during postnatal development and maturation of the central auditory system.

The formation of tertiary neurospheres has a significance in NSC identification from primary cultures. In the primary culture, multiple cell lineages may survive and constitute the spheres. The number of non-NSCs usually diminishes in 1–2 weeks in the suspension culture medium, whereas NSCs can form neurospheres for many passages. Therefore, after culturing for 1–2 weeks, the tertiary spheres are mainly composed of NSCs. In the in vitro study, AC tissues from P3, P14, and 2M were harvested for neurosphere study. The 4M group was excluded from the in vitro study because of the remarkably lower percentage of NSCs in vivo. It is found that AC cells of the P3 and P14 groups were able to form primary, secondary, and tertiary spheres. 2M mouse AC cells were able to form primary and secondary spheres, but tertiary spheres were rarely observed. Because the 2M group did not form distinct tertiary spheres, this group was excluded from the quantitative study.

In general, the neurosphere size is usually determined by the NSC sphere-forming ability, which includes the percentage and the proliferation ability of NSCs in the sphere. Therefore, the sphere-forming ability of NSCs, including the percentage and the proliferation of NSCs, can be indicated by the sphere size. In the sphere size quantitative study, it is observed that the sphere size was significantly larger in the P3 group. These data suggest that the sphere-forming capability decreases with age and that adult mouse AC has a limited sphere-forming ability. It is noted that sphere-forming ability is an ambiguous measurement, so cell proliferation assays were performed in this study.

Two complementary proliferation assays, including the cell doubling time and EdU incorporation tests, were applied to this study. Both assays suggest that the proliferation rate of the P3 AC group is significantly higher than the P14 group. In the NSC protein study, Nestin and Sox2 expression was observed in both P3 and P14 groups, and the percentage of Nestin and Sox2 double-positive cells of the P3 group was significantly higher than the P14 group. These data reveal that the proliferation and the Sox2/Nestin expression of the P3 groups are considerably better than those of the P14 group, suggesting that the neural stem cell features of AC NSCs decrease with age. These results are consistent with previous studies, which showed that the proliferation and neurogenesis ability of central nervous system NSCs reduced with age [17,36].

In this study, AC NSCs were investigated both in vitro and in vivo. The percentage of Nestin and Sox2 double-expressing cells decreases with age, both in vitro and in vivo. The percentage of Nestin and Sox2 double expressing cells are approximately 15%, 5%, 4%, and 2% for P3, P14, 2M, and 4M, respectively, in vivo, whereas these numbers are 81% and 42% in P3 and P14 tertiary neurospheres respectively in vitro. The higher percentage of double expressing cells in vitro indicates that double expressing NSCs contribute to the majority of sphere-forming cells. Although approximately 4% double expressing cells are observed in the 2M AC in vivo, the sphere-forming ability of these cells is very limited, which fails to form distinct tertiary spheres in vitro. The reason for the decreased sphere-forming ability of the 2M group may be related to the low percentage of NSCs as well as reduced proliferation ability [17,36].

In summary, AC tissues were studied for the presence of NSCs both in vitro and in vivo in this report. NSCs were observed in the postnatal and young adult AC area. During postnatal development, significantly reduced cell proliferation and NSC protein expression were observed. For the early postnatal stage mouse, the percentage of NSCs and the proliferation ability are relatively high, indicating that these local NSCs may be a valuable source that can be used for replacement therapy. As for the late postnatal stage and adult mouse, the low percentage of NSCs and relatively poor proliferation ability may restrict the application of these NSCs. These data may suggest that exogenous NSCs may be required for the mature AC cell replacement. Currently, there is a lack of AC NSC study in higher species. Future studies using higher species and potentially clinical human data may be required to further correlate the results of this study to the neurodevelopment and neurodegeneration of human AC. Additionally, the maturation and function of AC NSCs, as well as the integration of AC NSCs into the AC auditory pathway, should be investigated using both in vitro and in vivo models in the future. The outcomes of this study may provide cues for the development of stem cell-based therapy to replace the damaged cellular structures in the central auditory system in the future.

## 4. Materials and Methods

### 4.1. Animals

Wildtype Swiss Webster mice (Charles River) were used in this study. This study was carried out in accordance with the guidelines of the Wayne State University Institutional Animal Care and Use Committee. The protocol of animal use and care was approved by the above-mentioned committee, approval code: IACUC-17-11-0387, approval date: 30 January 2018.

### 4.2. AC Tissue Collection and Immunofluorescence

Swiss Webster postnatal day 3 (P3), postnatal day 14 (P14), 2-month-old (2M), and 4-month-old (4M) mice were euthanized, and their brain samples were harvested and fixed by 4% paraformaldehyde (n = 5 animals per group). The brain tissue was treated with 30% sucrose overnight, followed by cryosection at 10 µm thickness. The samples were treated with blocking solution containing 5% donkey serum, 0.2% Triton X-100, and 0.1M phosphate-buffered saline (PBS) for 30 min at room temperature, followed by incubating with primary antibodies at 4 °C overnight. The primary antibodies were Sox2 (1:200; R&D) and Nestin (1:400; Developmental studies hybridoma bank, DSHB, Iowa City, IA, USA). The samples were treated with secondary antibodies, including Alexa Fluor 488, 549, or 647 conjugated donkey anti-mouse or goat antibodies (1:500; Jackson Immunoresearch, West Grove, PA, USA). Hoechst 33342 (1:1000, Invitrogen, Waltham, MA, USA) was used as a universal cell-permeant nuclear counterstain. All antibodies have been tested in our previous report using vendor suggested positive and negative controls [26], which was used to determine whether examined cells were positive or negative in this study.

### 4.3. Generation of AC-Derived Neurospheres

AC tissues of P3, P14, and 2M mice were isolated, dissociated, and cultured as in the previous report [26]. Briefly, AC tissues were isolated and rinsed in cold PBS solution, treated with 0.025% Trypsin (Invitrogen) for 8 min at 37 °C, followed with mechanical trituration. Dissociated cells were cultured in the suspension medium containing DMEM/F12, 1% N2, 2% B27, epidermal growth factor (EGF; 20 ng/mL), fibroblast growth factor 2 (FGF-2; 20 ng/mL), and 0.1% penicillin-streptomycin (all from Invitrogen) in a 37 °C incubator supplied with 5% CO_2_ for 5–7 days in the primary culture. At the end of the primary culture, the cell clusters and spheres were treated with TrypLE (Invitrogen) for 3–5 min in a 37 °C water bath. Dissociated cells were passaged into the new suspension medium for approximately 5–7 days for the secondary neurosphere culture. The same method was applied to cultivate the cells in the tertiary AC sphere culture.

### 4.4. Proliferation Assays

The tertiary AC spheres were treated with 5-ethynyl-2-deoxyuridine (EdU, 200 ng/mL, Sigma) for 48 h as previously reported [28,29,34]. The spheres were fixed by 4% paraformaldehyde at room temperature for 10 min, followed by incubation in Click-iT reaction buffer, CuSO4, Alexa Fluor 555, and reaction buffer additive (Invitrogen) for 30 min. All nuclei were labeled with Hoechst 33342 (nuclei marker). The sphere samples were observed and imaged using Leica SPE confocal microscopy.

### 4.5. Cell Quantifications

Nuclei, Nestin, and Sox2 immunostaining images were captured to quantify the percentage of NSCs in the AC area of brain sections. The quantification method has been reported previously [26,28]. Briefly, the number of Nestin, Sox2, Nestin & Sox2, and Hoechst 33342 positive cells were counted by the cell count and particle analyses tools of the ImageJ software (NIH). The percentage of positive cells is equal to (the number of Nestin, Sox2, or both positive cells)/(the number of Hoechst 33342 positive cells) × 100%.

Tertiary AC sphere images were captured to quantify the diameter of AC spheres. The diameter of each sphere was measured by ImageJ software length measure tool as previously reported [28,30]. The cell number of the primary, secondary, and tertiary AC spheres of P3, P14 was determined to calculate the doubling time using previously reported methods [37,38]. The doubling time is equal to (t1 − t0)/log2(c1 − c0). c1 = the cell number at time point t1, and c0 = the cell number at time point t0.

Nuclei and EdU staining images were captured to quantify the percentage of proliferation cells in AC spheres. The method for the quantification was shown in our previous reports [28,29]. Briefly, the number of EdU and Hoechst 33342 positive cells were counted by the ImageJ software cell count tool. The percentage of positive cells is equal to (the number of EdU positive cells)/(the number of Hoechst 33342 positive cells) × 100%.

### 4.6. Statistical Analysis

A blind method was used for the immunofluorescence, quantification, and statistical analysis. In the in vivo study, brain samples were collected from 5 mice for each group. In the in vitro study, the AC tissues of 2–3 animals were used for each primary culture experiment, and 3 independent primary culture experiments were performed for this research. Student’s *t*-test was applied for the analysis of the two-group study, whereas analysis of variance (ANOVA) and Tukey post hoc tests were used for the study containing 3 or more groups. In this study, *p* < 0.05 was selected as the criteria of statistical significance.

## Figures and Tables

**Figure 1 ijms-22-01550-f001:**
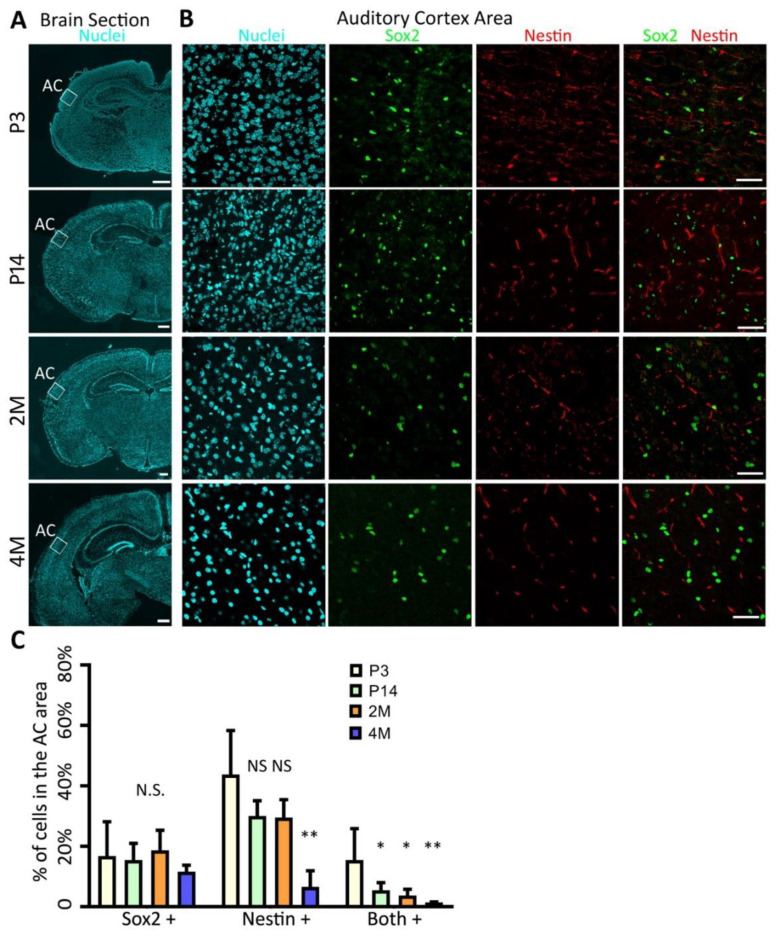
Presence of neural stem cells (NSCs) in the mouse auditory cortex (AC) in vivo. (**A**) Epifluorescence microscopy images show the mouse brain sections at postnatal day 3 (P3), postnatal day 14 (P14), 2 months old (2M), and 4 months old (4M), which is labeled with Hoechst 33342, a universal nuclei marker. White boxes indicate the AC area. (**B**) Confocal microscopy images show that a proportion of AC cells express NSC proteins Nestin and Sox2. (**C**) Quantification study of Nestin and Sox2 immunostaining shows the percentages of Nestin positive, Sox2 positive, and double-positive cells. The percentages of Sox2-expressing cells are similar among the 4 groups. However, the percentages of Nestin-expressing and double-expressing cells decrease with age. N.S., *, and ** indicate *p* > 0.05, *p* < 0.05, and *p* < 0.01 respectively. ANOVA followed by Tukey post hoc test; n = 5 animals/group. Scale bar: 500 µm in (**A**) and 50 µm in (**B**).

**Figure 2 ijms-22-01550-f002:**
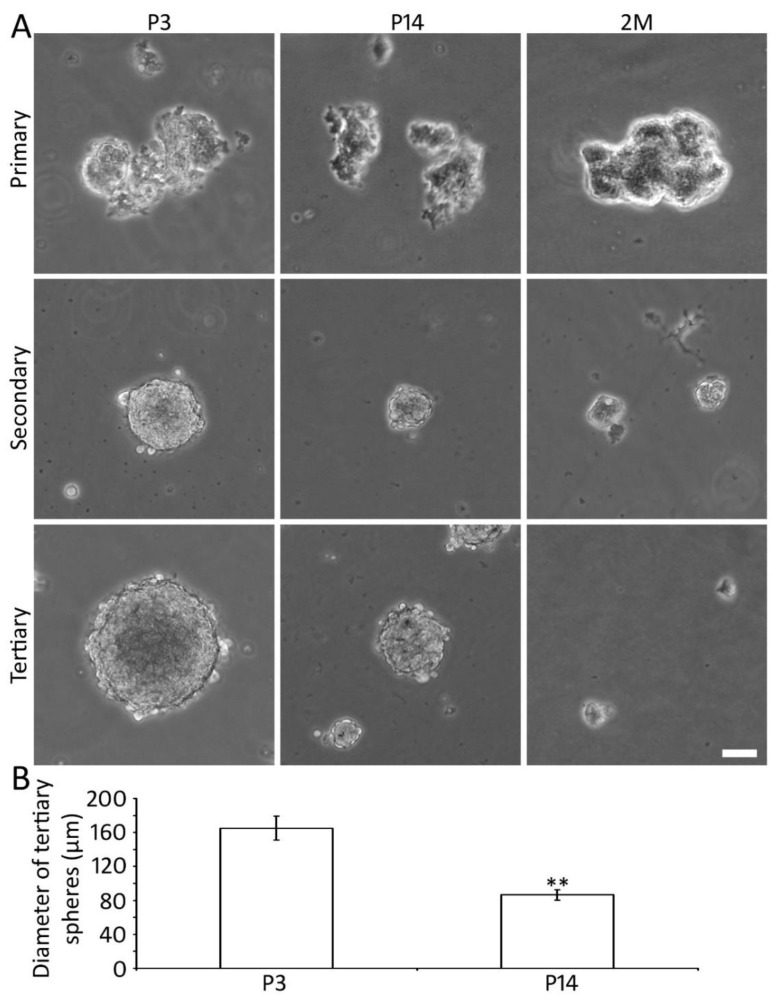
Sphere-forming ability of P3 and P14 mouse AC-derived cells in vitro. (**A**) Phase-contrast images show that dissociated P3, P14, and 2M AC cells form cell clusters and spherical structures when they are cultured in suspension in the primary as well as subsequent secondary and tertiary cultures. Cells of the P3 and P14 groups form distinct secondary and tertiary spheres, whereas cells of the 2M group fail to form distinct tertiary spheres. (**B**) Quantification study shows that the diameter of P3 AC-derived spheres is significantly larger than the P14 AC spheres (** indicated *p* < 0.01; Student’s *t*-test; n = 12 samples from 3 independent primary cultures). Scale bar: 50 µm in (**A**).

**Figure 3 ijms-22-01550-f003:**
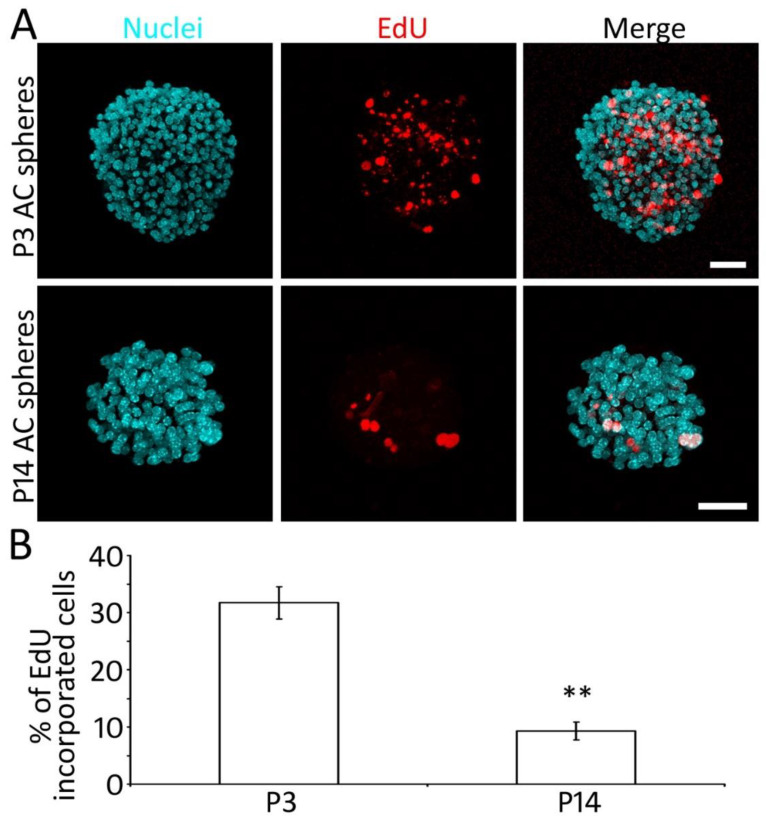
Proliferation of P3 and P14 AC-derived spheres. (**A**) Confocal microscopy images show that P3 and P14 AC-derived spheres incorporate EdU when EdU is added to the tertiary culture for 48 h. (**B**) Quantification study of the EdU incorporation shows that the percentages of EdU incorporated cells of P3 and P14 AC-derived spheres. The percentage of EdU incorporated cells of the P3 group is significantly higher than the P14 group (** *p* < 0.01, Student’s *t*-test, n = 10 samples from 3 independent primary cultures). Scale bar: 50 µm in (**A**).

**Figure 4 ijms-22-01550-f004:**
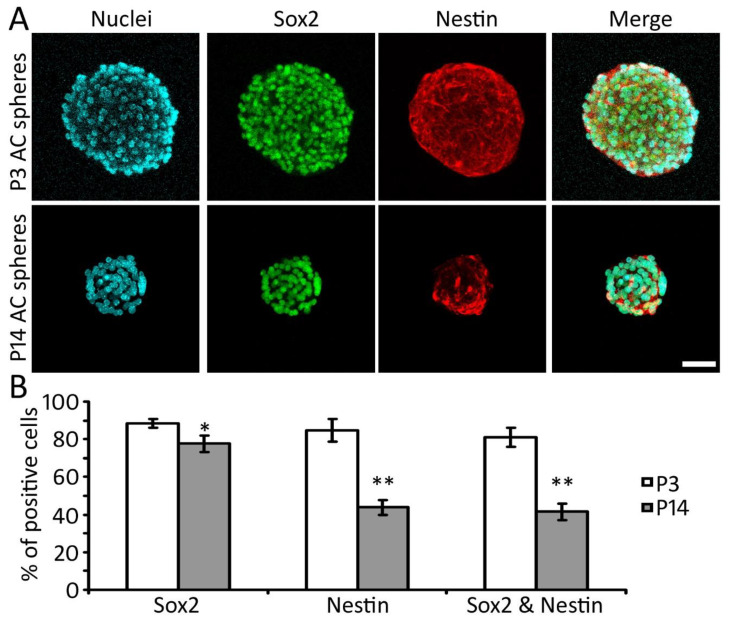
Characterization of AC-NSCs of P3 and P14 AC-derived spheres (**A**) Confocal microscopy images show that P3 and P14 AC-derived spheres express NSC proteins Sox2 and Nestin. (**B**) The quantification study of the P3 and P14 AC-derived sphere immunostaining shows that the percentages of Nestin positive, Sox2 positive, and double-positive cells of the P3 group are significantly higher than those of the P14 group (* and ** indicate *p* < 0.05 and *p* < 0.01, respectively, Student’s *t*-test, n = 10 samples from 3 independent primary cultures). Scale bar: 50 µm in (**A**).

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
