# Peer review of "Postnatal Changes of Neural Stem Cells in the Mammalian Auditory Cortex"

_ijms, 2021, doi:10.3390/ijms22041550_

Round 1

Reviewer 1 Report

Following up on previous work by this group, the manuscript by Hu et al., identifies and quantifies neural stem cell populations in the auditory cortex of mice aged p3, p14, 2M and 4M. These AC-NSCs appear to decrease in prevalence over the course of development in the measured time points, but are still viable as stem cells in vitro from the two youngest age groups. The manuscript is straightforward and sound, with some addressable concerns as follows:

  1. How many sections per animal? What was the N used for statistical analysis (animal, section, cell for the IHC experiments and animal, culture or sphere for the NSC experiments)?
  2. What criteria were used to determine whether a cell was positive or negative for a specific IHC marker and was this kept constant between animals/sections? Were observers blinded to group when quantifying?
  3. Could the authors discuss the significance of identification of tertiary (vs primary and secondary) neurospheres? Could the authors also discuss the relevance of the sphere size and sphere forming ability in general?
  4. Could the authors just briefly state in the manuscript whether any data were collected in the culture experiment for the 4M group and if no data were collected, why this time point was dropped?
  5. It is interesting that Sox2 expression, and to a lesser extent Nestin expression, seem stable across the assessed developmental time points, with only co-expression of these markers decreasing over the course of development. Can the authors discuss the use of Nestin and Sox2 as markers of neural stem cell populations, and perhaps hypothesize what the Nestin+/Sox2- and Nestin-/Sox2+ populations are if not NSCs?
  6. Are there data available in other higher species on NSCs in AC? Could the authors discuss the relevance of the selected time points in general and perhaps more specifically as it relates to human neurodevelopment of auditory cortex, and with relation to the pathophysiological conditions where NSC transplants may be useful?
  7. Follow up studies looking at functional maturation of these cells and perhaps integration into the broader AC network and/or whether cultured NSCs are viable AC xenografts would be interesting.

Minor:

  • The first sentence in the results, “AC-NSCs were identified in the postnatal mouse brain in vitro” is unclear, I might suggest rephrasing to indicate that stem cells were pulled from the postnatal brain and cultured in vitro?

Author Response

We appreciate all the constructive comments and suggestions from the reviewer, and we have adopted all the suggestions in our revised manuscript. Significant changes have been highlighted in the manuscript. The following are our point-to-point responses to reviewers’ comments:

“Following up on previous work by this group, the manuscript by Hu et al., identifies and quantifies neural stem cell populations in the auditory cortex of mice aged p3, p14, 2M and 4M. These AC-NSCs appear to decrease in prevalence over the course of development in the measured time points, but are still viable as stem cells in vitro from the two youngest age groups. The manuscript is straightforward and sound, with some addressable concerns as follows”:

1. How many sections per animal? What was the N used for statistical analysis (animal, section, cell for the IHC experiments and animal, culture or sphere for the NSC experiments)?

Response: In the in vivo study, brain samples were collected from 5 mice for each group, and approximately 20 sections were collected from each animal. In the in vitro study, the AC tissues of 2-3 animals were used for each primary culture experiment, and 3 independent primary culture experiments were performed for this research. The N indicates the animal number (n=5 mice) and independent primary cultures (n=3) for the in vivo and in vitro studies respectively. (Page 6)

2. What criteria were used to determine whether a cell was positive or negative for a specific IHC marker and was this kept constant between animals/sections? Were observers blinded to group when quantifying?

Response: All antibodies have been tested in our previous report using vendor suggested positive and negative controls [26], which was used to determine whether examined cells were positive or negative in this study. (Page 5)

A blind method was used for the immunofluorescence, quantification, and statistical analysis. (Page 6)

3. Could the authors discuss the significance of identification of tertiary (vs primary and secondary) neurospheres? Could the authors also discuss the relevance of the sphere size and sphere forming ability in general?

Response: The formation of tertiary neurospheres has a significance in NSC identification from primary cultures. In the primary culture, multiple cell lineages may survive and constitute the spheres. The number of non-NSCs usually diminishes in 1-2 weeks in the suspension culture medium, whereas NSCs can form neurospheres for many passages. Therefore, after culturing for 1-2 weeks, the tertiary spheres are mainly composed of NSCs. (Page 4)

In general, the neurosphere size is usually determined by the NSC sphere-forming ability, which includes the percentage and the proliferation ability of NSCs in the sphere. Therefore, the sphere-forming ability of NSCs, including the percentage and the proliferation of NSCs, can be indicated by the sphere size.  (Page 4)

4. Could the authors just briefly state in the manuscript whether any data were collected in the culture experiment for the 4M group and if no data were collected, why this time point was dropped?

Response: The 4M group was excluded from the in vitro study because of the remarkably lower percentage of NSCs in vivo (Page 4). Our study also showed that neurospheres are rarely found in the 4M group.

5. It is interesting that Sox2 expression, and to a lesser extent Nestin expression, seem stable across the assessed developmental time points, with only co-expression of these markers decreasing over the course of development. Can the authors discuss the use of Nestin and Sox2 as markers of neural stem cell populations, and perhaps hypothesize what the Nestin+/Sox2- and Nestin-/Sox2+ populations are if not NSCs?

Response: Nestin is an intermediate filament protein that is usually observed in the nervous tissue and NSCs [34, 35]. Sox2 is a transcription factor that plays an important role in maintaining the stem cell features of embryonic stem cells and neural stem cells. Sox2 also has an essential downstream role in the differentiation of specific neuron subtypes [27, 36]. Simultaneous expression of Nestin and Sox2 usually indicates the presence of NSCs [28, 30, 37]. (Page 3)

6. Are there data available in other higher species on NSCs in AC? Could the authors discuss the relevance of the selected time points in general and perhaps more specifically as it relates to human neurodevelopment of auditory cortex, and with relation to the pathophysiological conditions where NSC transplants may be useful?

Response: Currently, there is a lack of AC NSC study in higher species. Future studies using higher species and potentially clinical human data may be required to further correlate the results of this study to the neurodevelopment and neurodegeneration of human AC. (Page 5)

7. Follow up studies looking at functional maturation of these cells and perhaps integration into the broader AC network and/or whether cultured NSCs are viable AC xenografts would be interesting.

Response: We appreciate the review comments. Indeed, the maturation and function of AC NSCs as well as integration of AC NSCs into the AC auditory pathway should be investigated using both in vitro and in vivo models in the future  (Page 5)

 Minor:

The first sentence in the results, “AC-NSCs were identified in the postnatal mouse brain in vitro” is unclear, I might suggest rephrasing to indicate that stem cells were pulled from the postnatal brain and cultured in vitro?

Response: We thank the suggestion and have changed it to “Our previous reports showed that the AC tissue was harvested from the postnatal mouse brain, and AC-NSCs were identified in the culturing of these tissues”. (Page 2)

Reviewer 2 Report

In this paper about Postnatal changes of neural stem cells in the mammalian auditory cortex, the authors aim to demonstrate the presence of neural stem cells (NCS) in the auditory cortex during postnatal development of mice. This paper seems to be the natural continuation of their previous paper demonstrating the presence of the NCS at postnatal day 3 in vitro. The results are well presented and discussed. The manuscript is very clear and well written.

Unfortunately, it is not surprising that the capabilities of the NCS to proliferate and to go through neurogenesis processes decrease during development as previously demonstrated in other papers. 

Author Response

We thank the Reviewer for the helpful comments.